# Do the Green Credit Guidelines Affect Corporate Green Technology Innovation? Empirical Research from China

**DOI:** 10.3390/ijerph18041682

**Published:** 2021-02-10

**Authors:** Min Hong, Zhenghui Li, Benjamin Drakeford

**Affiliations:** 1School of Economics and Management, Hunan Institute of Technology, Hengyang 421002, China; hongmin@hnit.edu.cn; 2Guangzhou International Institute of Finance, Guangzhou University, Guangzhou 510006, China; 3Economics and Finance Subject Group, Portsmouth Business School, University of Portsmouth, Portsmouth PO1 3DE, UK; ben.drakeford@port.ac.uk

**Keywords:** green credit guidelines, corporate green technology innovation, heterogeneity

## Abstract

Green technology innovation is regarded as an important means to achieve sustainable development. Countries all over the world mainly implement green technology innovation policies from the aspects of environmental regulation and financing constraints. The effect of financing constraint policy on enterprise green technology innovation remains to be investigated. Based on the event of “green credit guidelines” issued by China Banking Regulatory Commission in 2012, this paper collects the panel data of China’s 2825 listed companies from 2007 to 2018, constructs a difference-in-difference model, and studies the impact of green credit guidelines on corporate green technology innovation and its mechanism. The empirical results show: First, green credit guidelines can promote corporate green technology innovation on the whole. Second, the mechanism of green credit on enterprise green technology innovation is identified. Green credit guidelines mainly limited green technology innovation through reducing debt financing, rather than through financing constraints. Third, the impact of green credit guidelines on green technology innovation is heterogeneous. Green credit guidelines have a significant effect on the green technology innovation of state-owned and large enterprises, but have no effect on the green technology innovation of non-state-owned and small ones.

## 1. Introduction

Green technology innovation is considered as an important means to achieve sustainable development. Although technological innovation has solved many problems faced by human beings, it has not effectively solved the problem of carbon emissions caused by human activities and global temperature rise, which also seriously restricts the sustainable development. Green technology innovation can reduce the cost of emission reduction by using better technology [1] and thus can reduce waste, reduce pollution, improve ecology, promote the construction of ecological civilization, and realize sustainable development [2]. As one of the main carbon emission entities, corporate motivation towards green innovation technology plays a key role in global sustainable development. In order to maximize their own economic benefits and achieve some social goals, various countries formulate different policies to guide or urge corporates to achieve energy conservation and emission reduction by improving the resource efficiency. Through the guidance of green technological innovation policy, corporates can improve their resource efficiency and achieve win–win development of social energy conservation and emission reduction [3,4].

The government generally implements green technology innovation policy from two aspects, namely environmental regulation and financing restriction. As for environmental regulation policy, there is no consistent conclusion. On the one hand, environmental regulation can stimulate corporate innovation to reduce the production costs and improve the quality of products. Porter and Vanderlinde [3] and Porter [5] believe that well-designed environmental regulation can stimulate corporate technological innovation, which is usually called the “weak Porter Hypothesis”. Scholars mainly use pollution abatement and control expenditures (PACE) to measure the degree of environmental regulation. In different countries (or regions), the role of environmental regulation is not alike [6]. Jaffe and Palmer [7] and Brunnermeier and Cohen [8] are the first to carry out empirical research in this field. They use the data of the American manufacturing industry to confirm that environmental regulation can promote corporate innovation to a certain extent. Ambec and Barla [9] believe that environmental regulation can encourage firms to adopt innovative behaviors that bring profit growth.Hamamoto [10] concluded that environmental regulation played a positive role in R&D expenditure of the Japanese manufacturing industry. Yang et al. [11] used the industry panel data of Taiwan, they also found that the degree of environmental regulation is positively correlated with the level of R&D investment. Rubashkina et al. [12] used the enterprise data of major European countries and found that environmental regulation has a significant role in promoting their innovation. In addition, other indicators are also used to measure the degree of environmental regulation. For example, Chakraborty and Chatterjee [13] studied the impact of Germany’s regulatory policy of banning “azo” dyes on upstream leather and textile firms in India, and found that environmental regulation promoted India’s innovation. Other scholars have also studied environmental technological innovation, for example, Horbach [14] found that environmental regulation, environmental management tools, and general organizational change can encourage corporate environmental innovation. Measuring environment regulation by environmental protection tax, Costa-Campi et al. [15] used the data of Spanish manufacturing industry to confirm the incentive effect of environmental regulation on enterprise R&D investment.

On the other hand, environmental regulation may bring additional governance costs to firms, which will reduce the resources available to corporates, thus reducing the level of technological innovation [16]. Whaller and Whitehead [17] argued that environmental regulation not only made corporates bear high costs, but also limited the flow of capital from promising innovation projects to pollution reduction projects, thus reducing the technological innovation ability of firms. Leonard [18] believed that compared with the firms less affected by environmental regulation, firms that were more affected by environmental regulation may lose their domestic and international market share. At the same time, they will also face the increasing operation and investment costs caused by strict environmental regulation. Therefore, they will tend to choose the regions with weak regulation to rearrange production and investment and reduce the share of innovation investment. Yuan and Xiang [19] used the data of Chinese manufacturing firms to draw the conclusion that increasing pollution control costs inhibited the innovation output of firms. Shi, et al. [20] estimated the impact of China’s carbon emissions trading pilot policy on corporate innovation output and concluded that this policy significantly inhibited the innovation of regulated and non-regulated firms. Kneller and Manderson [21] used the environmental protection expenditure to measure environmental regulation, and concluded that environmental regulation was not helpful in increasing the R&D investment in the British manufacturing industry, and explained that although environmental regulation increases environmental R&D investment, it has a crowding out effect on non-environmental R&D investment.

It is conductive to promote green technology innovation through financing constraints. Countries all over the world have issued a series of financing policies to encourage green technology innovation; for instance, in 1991, Poland established the environmental protection bank, which focuses on supporting investment projects to promote environmental protection. In 1993, in order to further promote the development of energy-saving technology, Japan’s Ministry of General Production increased the total amount of financial investment and loans related to energy and environment from 560 billion yen in 1992 to 970 billion yen. In 2012, the UK government invested 3 billion pounds to establish the first green investment bank, the UK Green Investment Bank, focusing on green infrastructure projects with commercial value. China’s green finance policy started late. China Banking Regulatory Commission issued the “green credit guidelines” in 2012 to guide the banking financial institutions to issue green credit, and would evaluate on the effectiveness of green credit in banking financial institutions. Differently from other green finance policies, China’s green credit guidelines is a policy issued by the national banking supervision department, which has a strong binding force.

There are many literatures on the policy effect evaluation of environmental regulation, but the policy effect evaluation based on financing constraints is relatively scattered. This paper studies the policy effect of financing constraints on green technology innovation. The main work includes the following aspects: (1) studying the effectiveness of green credit policy on green technology innovation. Green credit policy has a significant impact on corporate financing, but there is no consistent conclusion on corporate green technology innovation. This paper takes China’s “green credit guidelines” as a case to evaluate the effect of green credit policy on enterprise green technology innovation. The reason for choosing China’s green credit guidelines as a case is that China is an emerging market country, and big economies are different from small ones. The evaluation of green credit policy can not only examine the role of green credit on the enterprise development in a specific stage, but also examine the heterogeneous effect of green credit policy. These results show that the introduction of green credit policy can significantly promote green technology innovation. (2) This paper analyzes the mechanism of green credit on green technology innovation. The results show that green credit policy mainly reduces green technology innovation through the level of short-term debt and long-term debt financing, rather than through financing constraints. (3) This paper analyzes the heterogeneous effect of green credit policy on green technology innovation. To a certain extent, firms with different attributes will have different motives for green technology innovation, so that the impact of green credit guidelines on green technology innovation is heterogeneous. Property right structure leads to managers’ decision-making differences on strategic green technology innovation, and firm size determines its financing flexibility. To a much greater extent, green credit affects firms’ financing costs and ways. Therefore, the heterogeneous impact of green credit on firm’s green technology innovation is mainly reflected in the property right structure and firm size. The empirical results confirm the existence of this heterogeneity and show that green credit policy has a significant effect on state-owned and large firms, but has no effect on non-state-owned firms and small and medium-sized firms.

The rest of the article is organized as follows: Section 2 is the model design, variables and data sources, and benchmark regression analysis results. Section 3 analyzes the mechanism of green credit policy on green technology innovation. Section 4 further discusses the heterogeneous impact of green credit policy on firms with different property rights and firms with different sizes. Section 5 briefly describes limitations of the study. Finally, Section 6 concludes and puts forward relevant policy suggestions.

## 2. Econometric Test on Whether Green Credit Guidelines Affect Green Technology Innovation

### 2.1. Model Design

The impact of green credit policy on corporate green technology innovation is mainly realized through policy incentives. Specifically, firstly, green credit policy will attract public attention to green products, making firms have to face the supervision of the public, pay attention to their own social responsibility, and consciously improve their green technology innovation. Second, according to the “innovation compensation effect” in the Porter hypothesis, when faced with the green credit policy, in order to obtain more credit support or better loan interest rate, firms often improve environmental governance technology, promote cleaner production, or increase environmental governance to control environmental risks [10,22]. Finally, green credit policy can effectively reduce the uncertainty in the process of corporate R&D investment and then reduce the risk.

The higher level of green credit will encourage firms to consciously improve environmental governance technology and improve the level of green technology innovation; in contrast, with higher level of green technology innovation, it is easier for firms to obtain green credit funds from banks. Then, we can find that firms with a higher level of green credit will produce higher level of green technology innovation. In this way, there is mutual causality between the explanatory variable and the explained variable, which will make the explanatory variable related to the error term, leading to endogenous problems and thus inconsistent estimation of parameters [23]. When the policy is exogenous, the difference-in-difference method can effectively alleviate the endogenous problems [24]. Moreover, the difference-in-difference method can not only control the unobservable individual heterogeneity between samples, but also control the influence of unobservable time-varying population factors, so we can obtain the consistent estimator of policy effect. With these good properties, the difference-in-difference method has been widely used in the field of policy evaluation. Based on this, we use the event of “green credit guidelines” issued by China Banking Regulatory Commission on 24 February 2012 (before the promulgation of the “green credit guidelines” in 2012, the credit policies in banking financial institutions paid less attention to green development issues; while the “green credit guidelines” issued in 2012 required banking financial institutions to clarify the direction and key areas of green credit support, to formulate special credit guidelines for industries with major environmental and social risks and those are restricted by national regulation, and to implement differential and dynamic credit policy), and establish a difference-in-difference model to evaluate the impact of green credit guidelines on green technology innovation. The specific model is as follows:(1)$$\begin{array}{c} \mathrm{Greeninnovation}=\beta_{0}+\beta_{1}\mathrm{Post}+\beta_{2}\mathrm{Treatment}\;\mathrm{firms}+\beta_{3}\mathrm{Post}\times \mathrm{Treatment}\;\mathrm{firms}+\beta_{j}{\mathrm{Controls}}_{j}+\mathrm{Firmdum}\\ \;+\mathrm{Yeardum}+\varepsilon \end{array}$$

Among them, Greeninnovation refers to the number of green invention patent applications of enterprises (divided by the number of invention patent applications), Post is the event dummy variable, when the year is in 2012, otherwise the value is 0; Treatment firms is the group variable, whose value is 1 when firm is in treatment group, otherwise the value is 0 (as the enterprises in heavy pollution industries are directly affected by the green credit guidelines, they are treated as treatment group and non-heavy pollution industries as control group). ${\mathrm{Controls}}_{j}$ includes a series of firm-level control variables. $\beta_{3}$ is the coefficient on the interaction term, which measures the impact of green credit guidelines on green technology innovation. This paper also adds to variable Firmdum and Yeardum to capture the effects of not time-varying firm-level factors and time-related factors. Additionally, we also consider constructing the model with absorptive capacity included, because the absorptive capacity can affect the adoption process and the cost of green technology innovation [25]. Scholars usually use the number of patents [26] or R&D personnel [27] to measure absorptive capacity. Therefore, in order to ensure the robustness of the results, we also add the number of patents as the measure of absorptive capacity in the model, and the results are still valid (see Table A1 in the Appendix A). The coefficient of the interaction term is still significantly positive, indicating that green credit guidelines can promote corporate green technology innovation.

According to the theory of technology innovation, we select five main control variables: (1) Firm size. According to Schumpeter’s hypothesis, the larger the firms are, the more efficient technological innovation will be. In other words, large firms are more innovative than small firms [28]. (2) The nature of firms’ property rights. Scholars usually think that state-owned enterprises are faced with soft budget constraints, which is not conducive to technological innovation. For example, Hart et al. [29] and Shleifer [30] demonstrated that when the ownership is owned by the government, managers usually have no incentive to invest in innovation to reduce costs and improve quality. Therefore, private ownership is usually superior to government ownership when innovation incentive and cost reduction are needed. Qian and Xu [31] analyzed the hindering effect of bureaucracy on innovation in a centralized economy under a soft budget constraint framework. (3) Firm’s ownership structure. The relationship between ownership concentration and technological innovation is still controversial. On the one hand, some scholars believe that improving ownership concentration may be conducive to technological innovation. According to Shleifer and Vishny [32], the increase of shareholding ratio will make the majority shareholders strengthen the supervision of opportunistic behavior of managers, which can improve firms’ technological innovation. On the other hand, some scholars believe that the improvement of ownership concentration hinders technological innovation. Demsetz and Lehn [33] believe that when the shareholding ratio is high enough, majority shareholders are able to expropriate the minority shareholders by colluding with managers and controlling the company’s decisions, so as to reduce firms’ technological innovation, which is generally known as the “expropriation effect” of majority shareholders. (4) Shareholding ratio of institutional investors. Generally speaking, greater institutional ownership is associated with more innovation. According to the research of Aghion et al. [34], when institutional investors hold a high proportion of shares, managers will pay attention to improving the level of technological innovation in order to reduce personal occupational risk. (5) Firm’s profitability. R&D projects have the characteristics of long duration and uncertain income, which makes it difficult for firms to attract external investment in R&D projects [35,36]. Firms need to have certain financial resources and ability to support R&D projects, so profitability is very important to support R&D innovation [37]. Audretsch [38] found that firm’s profitability usually affects their innovation activities, and firms with higher profitability are more willing to carry out technological innovation.

### 2.2. Variables and Data

The dependent variable is the level of green innovation, which is indicated by the number of green invention patents applications divided by the number of invention patent applications. The independent variable is group dummy variable, event dummy variable, and the interaction between them; the control variables include (1) firm size, which is indicated by the natural logarithm of the total assets at the end of the year. Scholars usually use the indicator “total assets” or “the number of employees” to measure the firm size. Scherer (1965) argues that both indicators can be used to measure firm size, and when “the number of employees” is used as the measurement indicator for firm size, the results can better support Schumpeter hypothesis, while when “total assets” is used, the results cannot support the hypothesis. However, in China, manufacturing enterprises account for the vast majority of the total listed enterprises, which makes it more appropriate to use total assets to measure the firm size. (2) Institutional investors hold shares. Institutional investors are represented by the proportion of the shares held by institutional investors; (3) State ownership. (4) Profitability is calculated by dividing the net profit by the average balance of assets in the year. (5) The top ten shareholders (top-ten holders): the top ten shareholders are indicated by the proportion of a shares held by the top ten shareholders. Either the variable “the shareholding ratio of the largest shareholder” or “the top ten shareholders” is used to measure the ownership structure. As the variable “the shareholding ratio of the largest shareholder” has not passed the significance test, we analyze the effect of “the shareholding ratio of the top ten shareholders” instead. Furthermore, some scholars also examined the impact of corporate governance on green innovation, such as the proportion of independent directors and the number of directors. However, none of these variables have passed the significance test, so these variables are not reported in the table. The mediating variables include: (1) the level of short-term debt (ST debt), indicated by the current debt balance at the end of the year; (2) the level of long-term debt (LT debt), indicated by non-current debt balance at the end of the year; (3) the level of financing debt (FI debt), indicated by the interest bearing debt financing balance at the end of the year; (4) the level of financing constraints (FC): according to the method of Kaplan and Zingales [39], the KZ financing constraint index is synthesized by the company’s operating net cash flow, dividends, cash holdings, asset liability ratio, Tobin’s Q and other financial indicators. This method has been widely used in the measurement of financing constraint indicators [40]. Specifically, we construct the KZ index by following these steps: first, we select the basic indexes to synthetic KZ indicators. These basic indexes include operating net cash flow divided by total assets in the previous period, cash dividends divided by total assets in the previous period, cash holdings divided by total assets in the previous period, asset liability ratio, and Tobin’s Q. Then the weight needs to be determined. We mainly classify the whole sample according to whether these basic indexes are lower than the median. If they are lower than the median, kz1, kz2, kz3, kz4, and kz5 are taken as 1, otherwise 0, KZ index is calculated: KZ = kz1 + kz2 + kz3 + kz4 + kz5. Ordered logistic regression is used to estimate the weight of the indicators. The coefficients of the parameters are the weight. Finally, we can synthesize the KZ financing constraint index. The larger the value is, the higher the degree of financing constraints faced by listed companies.

We mainly selected the data of Chinese listed companies from 2007 to 2018. Since the Ministry of Finance revised the accounting standards for business enterprises in February 2006, the financial data before 2007 is different from that in and after 2007. Meanwhile, considering the availability of green invention patent data, we mainly choose the data from 2007 to 2018. There are four sources of data: first, the number of green invention patent applications mainly comes from the website of the State Intellectual Property Office (http://pss-system.cnipa.gov.cn/sipopublicsearch/portal/uiIndex.shtml, accessed on 12 January 2021). We mainly identified the green patent based on the International Patent Classification in the “green list of International Patent Classification” launched by the World Intellectual Property Organization (WIPO) in 2010 (https://www.wipo.int/classifications/ipc/green-inventory/home, accessed on 12 January 2021). Specifically, we collected the data in two steps. First, we retrieved the patents by the keywords of specific corporate name and year. Then we identified and counted the green patents according to the International Patent Classification. Second, invention patent applications come from China Royal Flush financial database. Third, the financial data such as bank loan balance, long-term loan balance, total assets at the end of the year, asset–net interest ratio, top-ten shareholding ratio, and the equity nature are mainly from China Security Market and Accounting Research (CSMAR) database; fourth, the indicators such as the shareholding ratio of state-owned equity and the shareholding ratio of institutional investors are mainly from China RESSET database.

We mainly divide the heavy pollution industry and non-heavy pollution industry in two steps: First, according to the relevant provisions of “the 12th Five Year Plan for air pollution control in key areas” approved by the State Council of China, special emission limits of air pollutants shall be implemented in six major industries including thermal power, steel, petrochemical, cement, nonferrous metals, chemical industry, and coal-fired boiler projects in key control areas. Therefore, the enterprises in the six major industries of thermal power, iron and steel, petrochemical, cement, nonferrous metals, and chemical industry are preliminarily identified as heavy pollution industries. Second, according to “the classification management name of environmental inspection industry in listed companies” issued by the Ministry of environmental protection of China in 2008 and “the industry classification guidance of listed companies” revised by China’s Securities Regulatory Commission in 2012, we define 15 industries as the heavy pollution industries corresponding to the above six industries. These industries include the pharmaceutical manufacturing industry; non-metallic mineral products industry; wine, beverage, and refined tea manufacturing industry; power and heat production and supply industry; petroleum processing industry; coking and nuclear fuel processing industry; non-ferrous metal smelting and calendaring industry; chemical raw materials and chemical products manufacturing industry; rubber and plastic products industry; chemical fiber manufacturing industry; gas production and supply industry; ferrous metal mining and dressing industry; nonferrous metal mining and dressing industry; paper making and paper products industry; coal mining and washing industry; and ferrous metal smelting and rolling processing industry, while other industries are defined as non-heavy pollution industries.

Specifically, the definition of main variables, measurement indicators, and data sources are shown in Table 1.

On the basis of the original data, data cleaning is carried out: First, considering the great differences among financial enterprises, ST enterprises (Special Treat enterprises) with general enterprises; these two types of enterprises are eliminated in the sample. The reasons are as follows: There are great differences between the accounting system for financial enterprises and the accounting standards for business enterprises, so the financial indicators are not comparable between financial enterprises and business enterprises. Additionally, the financial situation of ST enterprises is abnormal, which makes them significantly different from ordinary enterprises. Second, the samples with lots of missing values are deleted. Third, the samples with unreasonable data are deleted, such as the samples with Institutional Investors is larger than 1 (it is impossible for the shareholding ratio to be over 1). Table 2 presents summary statistics for the full sample.

Table 2 reports the summary statistics of the main variables. It can be seen that, on average, corporate green invention patents accounts for 4.31% of the total invention patents, indicating that green innovation has occupied a certain proportion of all the innovation, though still not high. However, the standard deviation is 0.166, which is larger than the average, indicating that most of the green invention patents come from a few high patent intensive corporates. In addition, it can be seen that, on average, the top ten shareholders hold 58.17% of the shares, which indicates that the ownership concentration of China’s listed companies is relatively high; the proportion of state-owned shares reaches 6.49%, and the standard deviation of the proportion of state-owned shares is 15.79, which is larger than the average. This may be due to the fact that state-owned shares are mainly concentrated in a few state-owned enterprises, and the proportion of state-owned shares in non-state-owned corporates is relatively low.

### 2.3. Empirical Analysis

It has been explained that the nature of industry, the nature of property rights, firm size, ownership concentration, the shareholding ratio of institutional investors and profitability may affect green innovation. According to these characteristics, we group enterprises into sub-samples. Specifically, firm size, ownership concentration, institutional investors’ shareholding ratio, and profitability are classified according to tri-sectional quantiles, which can be divided into large-sized enterprises, medium-sized enterprises, and small-sized enterprises. According to the nature of property rights, the samples can be divided into state-owned enterprises and non-state-owned enterprises. According to the nature of the industry, they are divided into treatment group and control group. Finally, we calculate out the statistics of green innovation for these sub-samples, which are shown in Table 3.

Table 3 presents the summary statistics of green innovation for the subsamples. Compared with the control group, note that green innovation in the treatment group is slightly higher than that in the control group. There appears to be significant green innovation differences among the enterprises with different size: The larger the enterprises are, the more green innovation they produce. The level of green innovation for large enterprises, medium enterprises, and small enterprises is 0.360, 0.255, and 0.197, respectively. In addition, there are differences in green innovation between state-owned enterprises and non-state-owned enterprises, and non-state-owned enterprises produce more green innovation than state-owned enterprises. Finally, it is worth noting whether there exist differences in green innovation among enterprises with different a shareholding proportion of institutional investors and enterprises with different profitability. These results show that enterprises with more institutional investors and higher profitability tend to produce more green innovation. Therefore, we can conclude that there may be heterogeneity in green innovation for enterprises with different size, property nature, institutional investor shareholding ratio, and profitability.

Before regression analysis, it is necessary to test the correlation coefficient of the main variables to identify whether there is serious multicollinearity between the explanatory variables. Table 4 shows the results of Pearson correlation coefficient matrix.

Table 4 shows that green innovation is significantly correlated with top-ten holders, institutional investors and size. Among them, top-ten holders, institutional investors and size are positively correlated with green innovation, while state ownership is negatively correlated with green innovation. In addition, although there is a certain correlation between the main explanatory variables, the degree of correlation is not high, so we believe that there is no serious multicollinearity between the explanatory variables.

Correlation coefficient matrix can only test the correlation between two explanatory variables, but cannot test the multicollinearity of more than two explanatory variables. Therefore, in order to make the regression results more robust, we adopt the method of stepwise regression analysis. The specific results are shown in Table 5.

First, Column 1 of Table 5 presents the difference-in-difference regressions only with basic explained variables such as group dummy variables, event dummy variables, interaction variables, firm dummies, and year dummies. Column 2 gradually add variables representing the ownership structure such as top-ten holders, institutional investors, and state ownership. Finally, two other control variables such as size and profitability are added in Column 3.

Table 5 shows that green credit policy can promote green technology innovation of enterprises. The coefficient on post is 0.0335, and it has passed the 1% significance level test, which shows that after the introduction of green credit guidelines, green technology innovation of enterprises has improved significantly. The coefficient on treat is insignificant, which shows that there are no significant differences between the enterprises in heavy pollution industry and those in non-heavy pollution industry. More importantly, the coefficients on the interaction term Post*Treat from column 1 to column 3 are significantly positive, which indicates that green credit guidelines have a significant role in promoting green technology innovation of enterprises. With other factors unchanged, the proportion of green invention patents of enterprises in heavy pollution industries increased by 0.0124–0.0130, increased by 28.77–30.16%. These results verify the Porter hypothesis, that is, appropriate environmental regulation can encourage enterprises to carry out technological innovation. This is also consistent with several studies about China; for example, Zhao and Sun [41] measured the local environmental regulation intensity from industrial wastes and daily life wastes, used the panel data of China’s pollution intensive corporations, and has proved the applicability of porter hypothesis in China. Li et al. [42] has constructed a theoretical model and proved the effectivities of green loan and government subsidy on promoting green innovation. Hong et al. [43] proved that the policy of mandatory information disclosure can promote green technology innovation. However, this paper measures the environmental regulation from the financial constraints.

In addition, the empirical results also show that the coefficients of top-ten holders in column 2 and column 3 are −0.0003 and −0.0004, respectively, and they pass the significance level tests of 5% and 1%, respectively. It shows that the higher the proportion of top ten shareholders, the lower the level of green technology innovation. It supports the hypothesis of “encroachment effect” of majority shareholders, that is, when the ownership concentration increases, majority shareholders may be able to collude with managers to appropriate minority shareholders. The coefficient on size is significantly positive. It means that the larger the enterprise size is, the higher the level of green technology innovation. It is consistent with the “Schumpeter hypothesis”, that is, the larger the size of the enterprise, the more efficient the technological innovation. Large enterprises are more innovative than small enterprises. In addition, the proportion of state ownership, institutional investors, and profitability has not passed the significance test. It shows that the proportion of state-owned shares, the proportion of institutional investors, and the improvement of profitability cannot promote the green technology innovation of enterprises.

Then we test the parallel trend hypothesis, which is the precondition of the difference-in-difference model. We mainly refer to the method of Bertrand and Mullainathan [44], and take the year before the introduction of the green credit guidelines as the benchmark year, and then add three variables ${\mathrm{Post}}^{2008}$, ${\mathrm{Post}}^{2009}$, and ${\mathrm{Post}}^{2010}$ for the pre-period and six variables ${\mathrm{Post}}^{2013}$, ${\;\mathrm{Post}}^{2014}$, ${\mathrm{Post}}^{2015}$, ${\mathrm{Post}}^{2016}$, ${\mathrm{Post}}^{2017}$, and ${\mathrm{Post}}^{2018}$ for the post-period. We then next interact these nine timing variables with treatment. The specific model is as follows:(2)$$\mathrm{Greeninnovation}=\beta_{0}+\beta_{1}\mathrm{Post}+\beta_{2}\mathrm{Treatment}+\overset{2018}{\underset{t=2008\;t\neq 2011}{\sum }}\beta_{t}{\mathrm{Post}}^{t}\times \mathrm{Treatment}+\beta_{j\;}{\mathrm{Controls}}_{j}+\varepsilon$$

Among them, 2011 is the reference year, and ${\mathrm{Post}}^{t}$ is the time dummy variable. When the time is in and after 2012, the value of ${\mathrm{Post}}^{t}$ is 1; otherwise it is 0. Treatment is a group dummy variable. When the enterprise is in the treatment group, the value is 1, otherwise the value is 0. We focus on the coefficient on the interaction ${\mathrm{Post}}^{t}\times \mathrm{Treatment}$, which reflects the dynamic effect of green credit guidelines on green technology innovation. In order to show the dynamic effect more intuitively, we plot the coefficient and the confidence interval of 5% significance level, as shown in Figure 1.

Figure 1 shows that the parallel trend hypothesis is met. Specifically, before the introduction of green credit guidelines, all coefficients of the interaction term are not significant, while after the introduction, the coefficients of the interaction term in the third, fifth, and sixth years are significantly positive. The former means that before the introduction there is no significant difference of green technology innovation between heavy polluting enterprises and non-heavy polluting enterprises. However, in the third to sixth years after the introduction, the differences come into being (this can be understood as lag effect, for it generally takes more than one year for patent application until its acceptance). Heavy polluting enterprises produce significantly more green innovation than non-heavy polluting ones. Therefore, we can identify these differences as the effect of the introduction of green credit guidelines, which further prove the robust of above research conclusion.

## 3. Impact Mechanism of Green Credit Guidelines on Green Technology Innovation

### 3.1. Model Design

As a kind of credit fund, green credit is bound to play an important role in enterprise R&D. On the one hand, from the perspective of capital transmission, green credit has the “green” attribute. It provides preferential credit interest rates and credit policies for some energy-saving and environmental protection products with low pollution and low consumption, correspondingly increases the credit cost of enterprises in heavy pollution industries, reduces their debt level, and guides the capital flow of financial market from enterprises in heavy pollution industries to enterprises in non-heavy pollution industries. On the other hand, from the perspective of the characteristics, apart from general investment projects, technological innovation projects have the characteristics of high failure rate, high risk, large capital investment, and long durance [45], often facing more serious information asymmetry [46], so that financing constraints have become the “roadblock” of enterprise innovation [47].

The previous benchmark regression results show that the green credit guidelines have a promotion effect on the green technology innovation for heavy polluting enterprises. However, the green credit guidelines may affect the level of debt financing and financing ability of enterprises and affect the green technology innovation of enterprises. This paper discusses the indirect effect of green credit guidelines on green technology innovation from two mechanisms. Among them, the first mechanism is the restrictive effect caused by the reduction of debt financing level. That is to say, the green credit guidelines may reduce the level of debt financing of enterprises, and then reduce the green technology innovation of enterprises, which is called the “restriction mechanism”. The second mechanism is the hindrance effect caused by the rise of financing constraints. That is to say, green credit guidelines may increase the financing constraints of heavy polluting enterprises, and then hinder their green technology innovation, which is called the “financing constraint” mechanism. We mainly use the recursive regression equation of mediation effect model to identify the existence of these two mechanisms [48]. The model is set as follows:(3)$$\mathrm{Greeninnovation}=\beta_{0}+\beta_{1}\mathrm{Post}+\beta_{2}\mathrm{Treatment}\;\mathrm{firms}+\beta_{3}\mathrm{Post}\times \mathrm{Treatment}\;\mathrm{firms}+\beta_{j}{\mathrm{Controls}}_{j}+\varepsilon$$
(4)$$\mathrm{Debt}=\beta_{0}+\beta_{1}\mathrm{Post}+\beta_{2}\mathrm{Treatment}\;\mathrm{firms}+\theta \mathrm{Post}\times \mathrm{Treatment}\;\mathrm{firms}+\beta_{j}{\mathrm{Controls}}_{j}+\varepsilon \;$$
(5)$$\mathrm{Greenninovation}=\beta_{0}+\beta_{1}\mathrm{Post}+\beta_{2}\mathrm{Treatment}\;\mathrm{firms}+\delta_{1}\mathrm{Post}\times \mathrm{Treatment}\;\mathrm{firms}+\lambda \mathrm{Debt}+\beta_{j}{\mathrm{Controls}}_{j}+\varepsilon$$

According to the mediating effect model, the first step is to estimate Equation (3) and test whether the promotion effect of green credit guidelines on technological innovation exists, if $\beta_{1}$ is significantly positive, indicating that the introduction of green credit guidelines does have a positive role in promoting green technology innovation, and the benchmark regression model has been tested in the above; the second step is to regress Equation (4) to examine the relationship between green credit guidelines and intermediary variables, that is, the relationship between the introduction of green credit guidelines and the level of financing constraints and debt financing, and the expected coefficient θ is positive and negative, respectively. The third step is to estimate Equation (5), if $\beta_{1}$, θ, and λ are all significant, and $\delta_{1}$ in the two mechanisms is greater than and less than $\beta_{1}$, respectively. If neither θ or λ is significant, the Sobel test is needed to test the mediating effect.

### 3.2. Empirical Analysis

In order to test the mediating role of debt financing in green credit guidelines and green technology innovation, this paper uses the mediating effect model for regression analysis. Considering the differences between short-term liabilities and long-term liabilities, interest bearing liabilities and non-interest bearing liabilities, three kinds of indicators are used to measure the level of debt financing: the first indictor is the short-term debt financing level, which is mainly expressed by the year-end balance of current liabilities; the second indicator is the long-term debt financing level, which is mainly expressed by the sum of the year-end balance of long-term loans, bonds payable, long-term accounts payable, and special accounts payable. The third indicator is the level of financing liabilities, which is also called interest bearing liabilities. It is mainly expressed by the sum of the year-end short-term loans, long-term loans, non-current liabilities expiring within one year, and the balance of bonds payable. The results are shown in Table 6.

Table 6 shows that green credit guidelines can reduce green technology innovation through reducing the level of corporate debt financing. The coefficients of the interaction term Post*Treat from column 1 to 3 are −0.04, −1.095, and −1.072, respectively, which have passed the significance tests of 10%, 1%, and 1%, respectively. It shows that the introduction of green credit guidelines can significantly inhibit all kinds of debt financing levels (including short-term debt financing level, long-term debt financing level, and financing debt level). The coefficients of the interaction term Post*Treat from column 4 to 6 are 0.009, 0.011, and 0.011, respectively, and all pass the 1% significance test. The coefficients on ST Debt in column 4 and LT Debt in column 5 are 0.004 and 0.00027 respectively, and they pass the significance test of 1% and 10%, respectively. It shows that green credit guidelines can reduce short-term debt financing and long-term debt financing, and thus reduce the green technology innovation. In column 6, the coefficient on FI Debt did not pass the significance test. Furthermore, the results of the Sobel test are not significant. Therefore, it can be concluded that the green credit guidelines cannot reduce green technology innovation by reducing the level of corporate financing liabilities. Specifically, with other factors unchanged, after the introduction of the green credit guidelines, the level of short-term debt and long-term debt decreased by 0.04 and 1.095, respectively. It causes the green innovation to decline about 0.00046 (0.04 × 0.004)~0.00077 (1.095 × 0.00027), which accounts for 3.68% of the total effect (0.0125).

Lower debt financing does not necessarily lead to financing constraints. However, R&D investment is vulnerable to financing constraints due to its high risk, large capital investment, and long durance. Therefore, we also use the mediating effect model to examine the existence of the “financing constraint” mechanism. The results are shown in Table 7.

Table 7 shows that although green credit guidelines are able to improve the financing constraints of enterprises, they cannot reduce the level of green technology innovation of enterprises. The coefficient on the interaction term Post*Treat in column 1 is significantly positive. It shows that the green credit guidelines improve the level of financing constraints of enterprises. The coefficient on the interaction term Post*Treat in column 2 is significantly positive. The coefficient on financial constraints (FC) is not significant. Furthermore, the p value of the Sobel test is 0.7155, which indicates that it does not pass the mediating effect test. The result shows that green credit guidelines cannot reduce the level of green technology innovation by improving the financing constraints of enterprises.

Compared the results in Table 6 and Table 7, it can be concluded that green credit guidelines mainly affect the green technological innovation through the “restriction mechanism” rather than “financing constraint” mechanism.

## 4. Further Discussion: Heterogeneity Analysis

### 4.1. Heterogeneity of Influence Degree

According to the nature of property rights and the firm size, we analyze the heterogeneous impact of green credit policy on green technology innovation. According to the nature of property rights, the enterprises are divided into state-owned enterprises (SOE) and non-state-owned enterprises (NSE). According to the firm size, the samples are divided into three quartiles, the largest one-third of the samples are divided into large firms, and the smallest one-third of the samples are divided into small firms. Using the difference–difference model, the parameter estimation results are shown in Table 8.

Table 8 shows that green credit guidelines have a heterogeneous impact on green innovation. Specifically, green credit guidelines can promote green technology innovation in the sub-samples of state-owned enterprises and large-scale enterprises, but have no effect on state-owned enterprises and small-sized enterprises. In the sub-samples regression of state-owned enterprises and non-state-owned enterprises, the coefficients on the interaction term are 0.0213 and 0.0036, respectively. The former passed the significance test of 1% level, while the latter does not pass the significance test. The results show that green credit guidelines could only affect green innovation in state-owned enterprises rather than non-state-owned enterprises. In the sub-samples regression of large enterprises and small enterprises, the coefficients on the interaction term are 0.0225 and 0.0047, respectively. The former passed the significance test of 1% level, while the latter did not pass the significance test. Therefore, we can believe that green credit guidelines could only affect green innovation in large enterprises rather than small enterprises.

Therefore, green credit guidelines have a heterogeneous impact on enterprises with different attributes. The promotion effects are mainly reflected in state-owned enterprises and large-scale enterprises. The reason may be that state-owned enterprises and large enterprises are more sensitive to the government policy, which makes them respond faster and greater once the policy is issued. At the same time, according to the Schumpeter hypothesis, innovative activities often take high fixed cost and low marginal cost. Large enterprises have the advantages of scale economy, which can dilute the fixed cost, so they are willing to invest more in innovative activities.

### 4.2. Heterogeneity of Influence Mechanism

Furthermore, as mentioned above, since the heterogeneity of influence degree has been confirmed, is there heterogeneity of influence mechanism? In order to explore the heterogeneity of influence mechanism, we firstly divide samples into state-owned enterprises and non-state-owned enterprises, large-size enterprises and small enterprises. Then we use the mediating effect model to analyze the mechanism in sub-samples. The results are shown in Table 9.

Table 9 shows that the influence mechanism on enterprises with different property rights is heterogeneous. Panel A of Table 9 shows that the coefficients on the interaction term Post*Treat in column 2 and column 3 are both significantly negative, while the coefficients on the interaction term in column 1 are insignificant. These results indicate that green credit guidelines can reduce the level of long-term debt and financing debt of state-owned enterprises rather than the level of short-term debt. The coefficients on the interaction term from column 4 to column 6 are all significantly positive. The coefficients on ST Debt in column 4 are significantly positive. The coefficients on FI Debt in column 6 is significantly positive. However, the coefficient on LT Debt in column 5 is insignificant. Furthermore, the mediating effect test shows that neither the mechanism of ST Debt or LT Debt as the mediators exist. Therefore, these results indicate that green credit guidelines can reduce green innovation of state-owned enterprises mainly by reducing the level of financing liabilities of state-owned enterprises, rather than reducing the level of short-term debt financing and long-term debt financing.

Panel B of Table 9 shows that all the coefficients on the interaction term Post*Treat from column 1 to column 3 are significantly negative. However, none of the coefficients on ST Debt in column 4, LT Debt in column 5, or FI Debt in column 6 are significant. These results indicate that although green credit guidelines can reduce the level of short-term debt, long-term debt, and financing debt, they cannot thus reduce green innovation.

Therefore, whether state-owned enterprises or non-state-owned enterprises, the first action mechanism is established, namely, green credit guidelines will reduce the level of debt financing in both state-owned enterprises or non-state-owned enterprises. The differences are: green credit guidelines can reduce the level of long-term debt and financing debt in state-owned enterprises, while they can reduce all debts in non-state-owned enterprises including short-term debt, long-term debt financing, and financing debt. This may be due to the fact that state-owned enterprises have the government as the backstage guarantee. Therefore, the short-term debt financing such as accounts payable and employee compensation payable are not affected, but the interest bearing debt financing such as short-term loans, long-term loans, bonds payable, and the long-term debt level such as long-term accounts payable have decreased significantly, thus affecting the green technological innovation of enterprises. However, the second step of the influence mechanism is established only in the state-owned enterprises, whose green innovation is affected by green credit guidelines through debt financing. Specifically, green credit guidelines can inhibit green technology innovation by reducing the financing debt level of large enterprises.

This may be due to the fact that large enterprises have enough assets that can be used as collateral, so the interest bearing debt financing such as short-term loans, long-term loans, bonds payable, and the long-term debt level such as long-term accounts payable will not be affected. However, green credit does have some effect on the development of enterprises, so their commercial credit level such as accounts payable and employee compensation payable will be reduced, and thus will affect their green technology innovation.

Furthermore, this paper distinguishes enterprises with different sizes and examines their heterogeneous influence mechanism. The results are shown in Table 10.

Table 10 shows that the influence mechanism on enterprises with different sizes is heterogeneous. Panel A of Table 10 shows that the coefficient on the interaction term Post*Treat in column 1 is significantly negative, while neither the coefficient on the interaction term in column 2 or column 3 is significant. These results indicate that green credit guidelines can only reduce the level of short-term debt in large enterprises rather than the level of long-term debt and financing debt. The coefficients on the interaction term from column 4 to column 6 are all significantly positive. The coefficients on ST Debt in column 4 are significantly positive. Neither the coefficient on LT Debt in column 5 nor FI Debt in column 6 is significant. The results indicate that the higher the short-term debt financing level of large enterprises, the more green technology innovation can be promoted, while the long-term debt financing level and financing debt level have no significant effect.

Panel B of Table 10 shows that both the coefficients on the interaction term Post*Treat on column 2 and column 3 are significantly negative, while the coefficient on the interaction term in column 1 is insignificant. These results show that green credit guidelines can significantly reduce the level of long-term debt level and financing debt in small enterprises, but cannot reduce the level of short-term debt. However, none of the coefficients on interaction term from column 4 to column 6 are significant. Neither the coefficient on LT Debt in column 5 nor FI Debt in column 6 is significant. Although the coefficient on ST Debt is significant positive, the p value of Sobel test is 0.343, which denies the existence of mediating effect. These results indicate that although green credit guidelines can reduce the level of long-term debt and financing debt, they cannot thus reduce green innovation.

Therefore, whether large enterprises or small enterprises, the first step of influence mechanism is established. Green credit guidelines will reduce their debt financing level. Large enterprises exhibit a decrease in short-term debt level, while small enterprises exhibit a decrease in long-term debt level and financing debt level. The second step of the influence mechanism is only established in large enterprises, whose green innovation is affected by green credit guidelines through debt financing. Specifically, green credit guidelines can inhibit green technology innovation by reducing the short-term debt level of large enterprises. This may be due to the fact that large enterprises have enough assets that can be used as collateral, so the interest bearing debt financing such as short-term loans, long-term loans, bonds payable, and the long-term debt level such as long-term accounts payable will not be affected. However, green credit does have some effect on the development of enterprises, so their commercial credit level such as accounts payable and employee compensation payable will be reduced and thus will affect their green technology innovation.

## 5. Limitations of the Study

This study has several limitations. First, although we argue that green credit guidelines can promote enterprise green technology innovation, and that the possible mechanism may be the policy incentive of green credit guidelines, there is no corresponding empirical test for this policy incentive mechanism. Second, due to lack of the corresponding data, we are unable to make a comparative analysis between one group of enterprises that have credit and another group of enterprises that do not have credit. We believe that if we finish these two tasks, the results will be more substantial and interesting. Thirdly, it is necessary to consider firm’s micro behavior from circular economy strategies. Especially, with the advent of the digital economy era, digital technologies are essential enables of the circular economy [49,50], and the effect of green credit guidelines on green technology innovation may show other heterogeneity characteristics. All of these limitations are the directions of our further study.

## 6. Conclusions

Based on the panel data of China’s A-share non-financial listed companies from 2007 to 2018, this paper constructs a difference-in-difference model to study the impact of green credit guidelines on corporate green technology innovation.

First of all, green credit guidelines have a significant role in promoting green technology innovation. The weak Porter hypothesis is still controversial about the effect of environmental regulation on corporate technological innovation. This study uses the samples of Chinese listed companies to empirically test that green credit guidelines can promote corporate green technological innovation. This positive effect may be due to the incentive effect of green credit guidelines, which is manifested in three aspects: (1) Green credit policy will attract public attention to green products, which makes enterprises have to face public supervision, attach importance to their own social responsibility, and consciously improve the level of green technology innovation. (2) Under the green credit policy, in order to obtain more credit support or better loan interest rate, enterprises often improve environmental governance technology, promote cleaner production, or increase environmental governance to control environmental risks. (3) Green credit policy can effectively reduce the uncertainty and risk in the process of R&D investment.

Secondly, the mechanism of green credit on corporate green technology innovation is identified. Green credit guidelines mainly limited green technology innovation through reducing debt financing, rather than through financing constraints. Specifically, green credit guidelines inhibit green technology innovation by reducing the short-term debt and long-term debt financing level.

Finally, the impact of green credit guidelines on green technology innovation is heterogeneous. The heterogeneity of green credit guidelines on corporate green technology innovation is manifested in the degree and mechanism of influence. On the one hand, the impact degree on different types of enterprises is heterogeneous. In the state-owned enterprises and large enterprises, green credit guidelines has a significant role in promoting green technology innovation, but in the non-state-owned enterprises and small enterprises, green credit guidelines has no significant impact on green technology innovation. On the other hand, the impact mechanism on different types of enterprises is heterogeneous. In large enterprises, green credit guidelines inhibit green technology innovation by reducing the level of short-term debt financing. In the state-owned enterprises, green credit guidelines can inhibit green technology innovation by reducing the level of financing liabilities.

## Figures and Tables

**Figure 1 ijerph-18-01682-f001:**
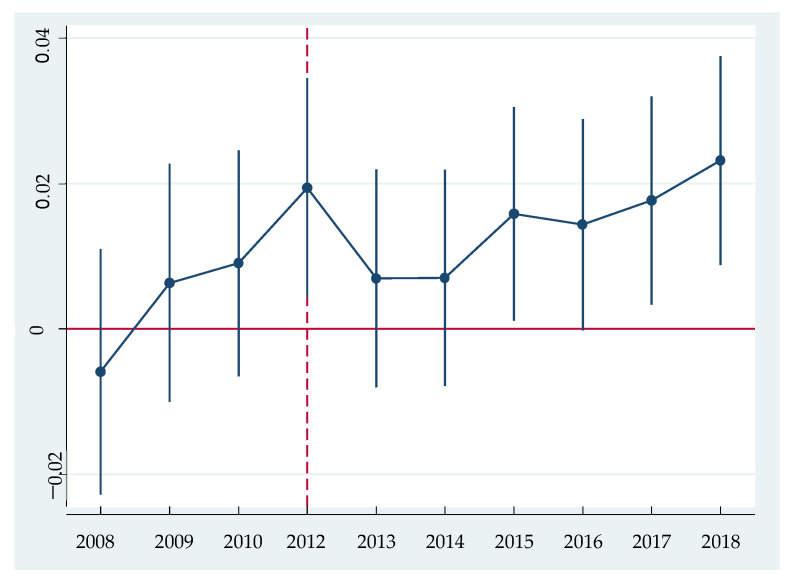
Parallel trend test results.

**Table 1 ijerph-18-01682-t001:** Main variables, measurement indicators, and data sources.

Variable	Meaning	Measurement Index	Data Sources
Green innovation	The proportion of green invent patent applications	Number of green patents divided by number of patents	Website of State Intellectual Property Office
Treatment	Group dummy variable	1 for heavy polluting enterprises and 0 for non-heavy polluting enterprises	Assignment by industry
Post	Event dummy variable	0 from 2007 to 2011 and 1 from 2012 to 2018	Direct assignment
Size	The size of the corporate	The natural logarithm of the total assets at the end of the year	CSMAR
State ownership	State ownership	Proportion of state-owned shares	RESSET
Institutional Investors	Institutional Investors	Shareholding ratio of institutional investors	CSMAR
Profitability	Profitability	Net profit divided by average balance of assets in the year	CSMAR
Top-ten holders	The proportion of the top-ten shareholders	Shareholding ratio of top ten shareholders	CSMAR
ST debt	Short-term debt	Balance of current liabilities at the end of the year	CSMAR
LT debt	Long-term debt	Balance of non-current liabilities at the end of the year	CSMAR
FI debt	Financing liabilities	Balance of financing liabilities at the end of the year	CSMAR
FC	Financial constraints	KZ index	Calculated

**Table 3 ijerph-18-01682-t003:** Summary statistics of green innovation for the subsamples.

Variable	Obs	Mean	Std.Dev.	Min	Max
Treatment group	8720	0.0442	0.169	0	1
Control group	14,103	0.0424	0.165	0	1
State-owned	9640	0.0404	0.158	0	1
Non-state-owned	13,183	0.0450	0.172	0	1
Large size	7609	0.360	0.272	0	0.993
Medium size	7605	0.255	0.227	0	0.996
Small size	7609	0.197	0.205	0	0.990
High institutional investors	7608	0.049	0.172	0	1
Medium institutional investors	7607	0.047	0.177	0	1
Low institutional investors	7608	0.033	0.148	0	1
High Top-ten holders	7608	0.042	0.168	0	1
Medium Top-ten holders	7606	0.047	0.172	0	1
Low Top-ten holders	7609	0.040	0.160	0	1
High profitability	7609	0.049	0.180	0	1
Medium profitability	7607	0.043	0.166	0	1
Low profitability	7607	0.037	0.152	0	1

**Table 4 ijerph-18-01682-t004:** Correlation coefficient test results of main variables.

	Variable	(1)	(2)	(3)	(4)	(5)	(6)
(1)	Green Innovation	1					
(2)	Top-ten Holders	0.015 **	1				
(3)	State Ownership	−0.033 ***	0.158 ***	1			
(4)	Institutional Investors	0.035 ***	0.255 ***	0.006	1		
(5)	Size	0.082 ***	0.119 ***	0.098 ***	0.308 ***	1	
(6)	Profitability	−0.002	−0.003	−0.003	−0.007	−0.056 ***	1

Note: ** and *** indicate significance at the levels of 5%, and 1%, respectively.

**Table 5 ijerph-18-01682-t005:** Empirical test on the impact of green credit guidelines on corporate green innovation.

Variable	(1)	(2)	(3)
Post	0.0335 ***	0.0316 ***	0.0249 ***
	(0.005)	(0.006)	(0.007)
Treat	−0.0045	−0.0041	−0.0035
	(0.007)	(0.007)	(0.007)
Post*Treat	0.0130 ***	0.0124 ***	0.0125 ***
	(0.004)	(0.004)	(0.004)
Top-ten Holders		−0.0003 **	−0.0004 ***
		(0.000)	(0.000)
State Ownership		−0.0000	−0.0000
		(0.000)	(0.000)
Institutional Investors		0.0008	0.0007
		(0.007)	(0.007)
Size			0.0039 **
			(0.002)
Profitability			0.0000
			(0.000)
Year FE	Yes	Yes	Yes
Firm FE	Yes	Yes	Yes
N	22,829	22,829	22,829
R^2^	0.007	0.007	0.007

Note: the brackets are standard errors, ** and *** at the top right of data indicate significance at the levels of 5%, and 1%, respectively.

**Table 6 ijerph-18-01682-t006:** An empirical test of green credit guidelines, financing level, and green technology innovation.

Variable	(1)	(2)	(3)	(4)	(5)	(6)
	ST Debt	LT Debt	FI Debt	Innovation	Innovation	Innovation
Post	0.730 ***	2.134 ***	1.536 ***	0.009 ***	0.011 ***	0.011 ***
	(0.015)	(0.133)	(0.106)	(0.003)	(0.003)	(0.003)
Treat	−0.241 ***	0.152	0.270	−0.004	−0.005	−0.005
	(0.037)	(0.316)	(0.251)	(0.007)	(0.007)	(0.007)
Post*Treat	−0.040 *	−1.095 ***	−1.072 ***	0.013 ***	0.014 ***	0.014 ***
	(0.022)	(0.192)	(0.153)	(0.004)	(0.004)	(0.004)
ST Debt				0.004 ***		
				(0.001)		
LT Debt					0.00027 *	
					(0.000)	
FI Debt						0.000
						(0.000)
Control variables	Yes	Yes	Yes	Yes	Yes	Yes
Firm FE	Yes	Yes	Yes	Yes	Yes	Yes
N	21,497	21,488	21,498	23,080	23,071	23,081
R^2^	0.239	0.025	0.014	0.006	0.006	0.006

Note: the brackets are standard errors, * and *** at the top right of data indicate significance at the levels of 10% and 1%, respectively; Columns (1)–(3) are regression results of formula (4), reflecting the effect of green credit policy on intermediary variables such as ST Debt, LT Debt, and FI Debt; Columns (4)–(6) are regression results of formula (5), reflecting the effect of green credit policy and intermediary variables on enterprise green technology innovation. ST Debt, LT Debt, and FI Debt indicate the level of short-term debt, long-term debt, and financing debt, respectively.

**Table 7 ijerph-18-01682-t007:** An empirical test of green credit guidelines, financing constraints, and green technology innovation.

Variable	(1)	(2)
	FC	Innovation
Post	−0.018 ***	0.012 ***
	(0.007)	(0.004)
Treat	−0.066 ***	−0.006 **
	(0.019)	(0.010)
Post*Treat	0.028 ***	0.012 ***
	(0.011)	(0.005)
FC		−0.000
		(0.004)
Control variables	Yes	Yes
Firm FE	Yes	Yes
N	17,211	17,211
R^2^	0.0406	0.0053

Note: The brackets are standard errors, ** and *** at the top right of data indicate significance at the levels of 5% and 1%, respectively; Column (1) is the regression result of Equation (4), reflecting the effect of green credit policy on intermediary variables; Column (2) is the regression result of Equation (5), reflecting the effect of green credit policy and intermediary variables on enterprise green technology innovation.

**Table 8 ijerph-18-01682-t008:** Heterogeneous impact of green credit guidelines on green innovation: sub-sample regression.

Variable	(1)	(2)	(3)	(4)
	SOE	NSE	Large Firms	Small Firms
Post	0.0161 *	0.0333 ***	0.0465 ***	0.0352 ***
	(0.009)	(0.011)	(0.013)	(0.010)
Treat	−0.0069	0.0008	−0.0146	−0.0164 **
	(0.010)	(0.011)	(0.017)	(0.008)
Post*Treat	0.0213 ***	0.0036	0.0225 **	0.0047
	(0.006)	(0.007)	(0.009)	(0.007)
Control variables	Yes	Yes	Yes	Yes
Year FE	Yes	Yes	Yes	Yes
Firm FE	Yes	Yes	Yes	Yes
N	9646	13,183	7610	7609
R^2^	0.0112	0.0065	0.0126	0.0064

Note: The brackets are standard errors, *, **, and *** at the top right of data indicate significance at the levels of 10%, 5%, and 1%, respectively.

**Table 9 ijerph-18-01682-t009:** Analysis on the influence mechanism in state-owned enterprises and non-state-owned enterprises.

Variable	(1)	(2)	(3)	(4)	(5)	(6)
	ST Debt	LT Debt	FI Debt	Innovation	Innovation	Innovation
**Panel A: The Mechanism on State-Owned Enterprises**						
Post*Treat	−0.012	−1.040 ***	−0.720 ***	0.022 ***	0.022 ***	0.022 ***
	(0.025)	(0.228)	(0.175)	(0.006)	(0.006)	(0.006)
ST debt				0.005 *		
				(0.003)		
LT debt					0.000	
					(0.000)	
FI debt						0.001 **
						(0.000)
Firm FE	Yes	Yes	Yes	Yes	Yes	Yes
N	9646	9646	9646	9646	9646	9646
R^2^	0.306	0.022	0.008	0.008	0.008	0.008
**Panel B: The Mechanism on Non-State-Owned Enterprises**						
Post*Treat	−0.077 **	−1.090 ***	−1.450 ***	0.005	0.005	0.005
	(0.034)	(0.306)	(0.245)	(0.007)	(0.007)	(0.007)
ST Debt				0.003		
				(0.002)		
LT Debt					0.000	
					(0.000)	
FI Debt						0.000
						(0.000)
Control variables	Yes	Yes	Yes	Yes	Yes	Yes
Firm FE	Yes	Yes	Yes	Yes	Yes	Yes
N	13,183	13,183	13,183	13,183	13,183	13,183
R^2^	0.274	0.039	0.039	0.005	0.005	0.005

Note: The brackets are standard errors, *, **, and *** at the top right of data indicate significance at the levels of 10%, 5%, and 1%, respectively.

**Table 10 ijerph-18-01682-t010:** Analysis on the influence mechanism in large enterprises and small enterprises.

Variable	(1)	(2)	(3)	(4)	(5)	(6)
	ST Debt	LT Debt	FI Debt	Innovation	Innovation	Innovation
**Panel A: The Influence Mechanism on Large Enterprises**						
Post*Treat	−0.053 *	−0.344	−0.185	0.024 ***	0.024 ***	0.024 ***
	(0.027)	(0.242)	(0.154)	(0.009)	(0.009)	(0.009)
ST debt				0.010 **		
				(0.004)		
LT debt					0.000	
					(0.000)	
FI debt						0.000
						(0.001)
Firm FE	Yes	Yes	Yes	Yes	Yes	Yes
N	7610	7610	7610	7610	7610	7610
R^2^	0.334	0.017	0.035	0.008	0.007	0.007
**Panel B: The Influence Mechanism on Small Enterprises**						
Post*Treat	−0.001	−1.415 ***	−1.062 ***	0.011	0.011	0.011
	(0.036)	(0.387)	(0.369)	(0.008)	(0.008)	(0.008)
ST Debt				0.007 **		
				(0.003)		
LT Debt					−0.000	
					(0.000)	
FI Debt						0.000
						(0.000)
Control variables	Yes	Yes	Yes	Yes	Yes	Yes
Firm FE	Yes	Yes	Yes	Yes	Yes	Yes
N	7609	7609	7609	7609	7609	7609
R^2^	0.076	0.007	0.012	0.006	0.006	0.006

Note: The brackets are standard errors, *, **, and *** at the top right of data indicate significance at the levels of 10%, 5%, and 1%, respectively. For the sake of brevity, the coefficients of post and treat are omitted in the table. Panel A and Panel B are the results of mechanism analysis of large enterprises and small enterprises, respectively, in which (1)–(3) are the first step of influence mechanism, that is, the effect of green credit policy on debt financing level, and (4)–(6) are the second step of influence mechanism, that is, the effect of green credit policy and debt financing level on green technology innovation.

**Table 2 ijerph-18-01682-t002:** Summary statistics.

Variable	Obs	Mean	Std.Dev.	Min	Max
Year	22,823	2013	3.390	2007	2018
Green innovation	22,823	0.0431	0.166	0	1
Largest holder	22,823	35.34	15.00	0.290	89.99
Top-ten holders	22,823	58.17	15.50	1.320	100.0
Institutional investors	22,823	0.271	0.246	0	0.996
State ownership	22,823	6.492	15.79	0	92.19
Size	22,823	21.92	1.329	10.84	28.52
Profitability	22,823	1.102	155.7	−51.95	23,510

## Data Availability

Data available in a publicly accessible repositories of China’s State Intellectual Property Office and three chargeable database such as China Royal Flush financial database, China Security Market and Accounting Research (CSMAR) database and China RESSET database.

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
