# Peer review of "Do the Green Credit Guidelines Affect Corporate Green Technology Innovation? Empirical Research from China"

_ijerph, 2021, doi:10.3390/ijerph18041682_

Round 1

Reviewer 1 Report

The authors construct a DID model to study the impact of green credit guidelines on corporate green technology innovation
based on the panel data of China's A-share non-financial listed companies from 2007 to 2018.
The analytical framework of DID is persuasive and the results are interesting.

Minor points:
・There are many Chinese characters e.g. P1 line 44, P2 line 49.
・The asterisks in Note of Table 4 is something wrong.

Author Response

Thanks very much for your comments and suggestions.We have corrected the mistakes in the paper.

Reviewer 2 Report

The paper is very nicely written and presents a relevant and timely topic, that of green technology innovation. The authors do a good job in framing the issue and the current state of knowledge. The introduction though is a bit long and could be shortened. I would also argue that it is important to mention some things about the role of circular economy strategies in this context. This is a growing phenomenon in companies and it is important to say a few things about how technology is influencing that. There is also some very interesting work which you should take into account on this topic by Kristoffersen et al., 2019 & 2020. It would strengthen the manuscript if you developed a bit of the implications in the conclusion section and talked about these papers in relation to your work.

Kristoffersen, E., Aremu, O. O., Blomsma, F., Mikalef, P., & Li, J. (2019, September). Exploring the Relationship Between Data Science and Circular Economy: An Enhanced CRISP-DM Process Model. In Conference on e-Business, e-Services and e-Society (pp. 177-189). Springer, Cham.

Kristoffersen, E., Blomsma, F., Mikalef, P., & Li, J. (2020). The smart circular economy: A digital-enabled circular strategies framework for manufacturing companies. Journal of Business Research120, 241-261.

Author Response

Thanks very much for your comments and suggestions. Please see the attachment to check the modification.

Reviewer 3 Report

The topic of Green Technology Innovation is on the agenda of multiple lines of research. The theme represents a huge challenge for the sustainable development of communities and activities at a global level, so it is pertinent and current.

The Abstract clearly shows the different phases of the study and the main conclusions. It would have been pertinent for the authors to mention how many companies are involved in this study, given that only “the companies listed in China” between 2007 and 2018 are mentioned.

The introduction has a sufficient literature review on the problem under study and is supported by different case studies from different countries. However, the comparison between the different policies of each country must be analyzed with scientific rigor. In the conclusions made by the authors in this section, an analysis of what is comparable and non-comparable in relation to the green credit policy would have been pertinent.

In line 144, the authors propose model design and justify their variables with other authors, however, it would have been interesting to compare them with other models to arrive at the specific model. When we build models it is important to apply theories that consider not only internal knowledge management processes, but also external knowledge exchange. For example, the absorptive capacity has been suggested by researchers as a concept that links knowledge generated outside the company to knowledge generated within the company.

In line 221, the authors characterize the variables and date, again it would have been pertinent to justify each variable in the light of other authors, so that the final results could be discussed and compared with the literature review. In the treatment of data and just as a suggestion, it would have been interesting for the authors to have chosen a group of financed companies as opposed to the chosen non-financed companies, to make a comparison.

In line 317, the authors present the empirical analysis, in which the differences-in-differences (DD) model is the most appropriate method for this type of studies. Once again, it would have been interesting to compare the results with that of other authors.

In general, the methodology followed in the study shows contributions to theory and practice. However, the variables of the models built lack a deeper explanation in confronting other authors. The authors should compare the models built with other existing models so that the results achieved are properly discussed. Selected companies should include two types: funded and unfunded.

Author Response

 I want to express my great thanks to you for your comments and suggestions. It is very helpful for us to revise the paper.

Reviewer 4 Report

The paper deals with a very interesting topic and provides a good basis for interesting future research. The paper is original, written in good English, with appropriate methodology and theoretical background as well as a good literature review.  The structure of the paper is logical, text is easy to read. The findings are presented and discussed. The aim of the paper was to study the policy effect of financing constraints on green technology innovation. The main work includes the following aspects: (1) study the effectiveness of Green credit policy on green technology innovation, (2) analysis of the mechanism of green credit on green technology innovation, (3) analysis the heterogeneous effect of green credit policy on green technology innovation. I like the scientific content of the paper, however, what I miss in the paper - is the section about the limitations of the study - authors should well describe it. Please add the section about limitations. 

Please correct also in line 49:  Porter and Vanderlinde [3] 与 Porter - delete Chinese and put "and" after [3] and check again the paper for all typos.

Author Response

Thanks very much for your comments and suggestions.And please see the attachment for the modification.
